# STATISTICAL VERIFICATION OF GENERAL PERTURBATIONS BY GAUSSIAN SMOOTHING

## ABSTRACT

We present a novel statistical certification method that generalizes prior work based on smoothing to handle richer perturbations. Concretely, our method produces a provable classifier which can establish statistical robustness against geometric perturbations (e.g., rotations, translations) as well as volume changes and pitch shifts on audio data. The generalization is non-trivial and requires careful handling of operations such as interpolation. Our method is agnostic to the choice of classifier and scales to modern architectures such as ResNet-50 on ImageNet.

## 1 INTRODUCTION

The success of deep neural networks (Krizhevsky et al., 2012; Silver et al., 2017) and their use in various application domains has triggered a concern to the sensitivity of these models to small, imperceptible perturbations, known as adversarial examples (Szegedy et al., 2014). Over the last few years there has been substantial interest in finding adversarial examples (Carlini & Wagner, 2017), empirically (Madry et al., 2018) and provably (Wong & Kolter, 2018; Mirman et al., 2018) defending against them as well as proving the neural network is robust to these (Gehr et al., 2018). Much of the focus so far has been on restricted, norm-based perturbations and while these cover important attacker models, it has been shown that natural perturbations, e.g., image rotation, can trigger adversarial behaviors not covered by norm-based attacks (Engstrom et al., 2017).

In this work we focus on certifying a neural network against a wider class of perturbations, by generalizing a recently presented statistical technique (Lecuyer et al., 2018; Cohen et al., 2019), called smoothing, for creating classifiers that are provably robust against certain norm-based attacks. The basic idea behind smoothing is to sample many possible perturbations of the input, classify each of them, and use the aggregate information to make a final robust classification. Overall, the method provides statistical guarantees for $l_2$-norm robustness around a given input (e.g., an image). In our work, we generalize this method to also handle perturbations of the *parameters* of a given transformation (e.g., the angle of a rotation). This generalization means that we can now apply and study the effectiveness of smoothing across a wider range of interesting perturbations (e.g., image rotation and translation), not previously possible. The generalization is non-trivial and requires careful handling of operations such as interpolation.

Our main contributions are:

- A generalization of the Gaussian smoothing framework to a richer class of perturbations, with a local robustness guarantee on the parameters of the perturbation.

- The first statistical verifier able to certify rotations and translations of images. The system scales to large input sizes and networks such as ImageNet classification. Concretely – among other results – we are the first to create *and* certify a provable ImageNet classifier (with high confidence) for image translation and rotation.

- An evaluation showing the strengths and limitations of the smoothing approach across a richer class of perturbations than previously possible.

## 2 RELATED WORK

We now survey the most closely related work in exact and statistical certification as well as richer perturbations beyond norm-based attacks.

**Exact Certification: complete and incomplete methods**  To defend against $l_p$ norm bound adversarial examples, many techniques have been developed to verify that a classification is stable in the presence of an attacker. These techniques include complete methods which guarantee a proof of robustness or provide a counter example, such as SMT solvers (Ehlers, 2017; Katz et al., 2017; Bunel et al., 2018), as well as incomplete methods which are sound but may suffer from false positives due to too much approximation error. Examples of these include abstract interpretation based methods (Gehr et al., 2018; Gowal et al., 2018; Singh et al., 2019), linear relaxations (Zhang et al., 2018; Weng et al., 2018) and semi definite programming (Raghunathan et al., 2018). Kurakin et al. (2017); Madry et al. (2018) train empirically robust networks by including adversarial examples in the training set. Wang et al. (2018a;b) replace the softmax layer to increase adversarial robustness. While both methods improve empirical robustness, the method does not provide a formal certificate. Provable defenses (which use incomplete methods) aim to address the issue to some degree by training networks in a way where they are more provable (Mirman et al., 2018; Wong & Kolter, 2018). However, because complete methods are not scalable and incomplete methods lose precision (by design), exact formal certification and training of large-scale networks with good absolute guarantees is a challenging task and remains an active area of research.

**Statistical Certification via Smoothing**  One can trade the absolute (exact) guarantees for probabilistic ones by using smoothed classifiers that scale to much larger networks (Lecuyer et al., 2018; Li et al., 2018; Cohen et al., 2019; Salman et al., 2019). A smoothed classifier $g$ can be built out of an ordinary classifier $f$ by taking the majority vote among perturbed inputs $x$, i.e., $f(x + \delta)$ where $\delta$ is drawn from a probability distribution. Smoothing has the advantage that it scales to large models, however, it can suffer from an added overhead during inference time, and currently only provides $l_2$ robustness statistical certification on limited input perturbations (e.g., pixels of an image). In this space, Lecuyer et al. (2018) presented the first certified robustness method based on randomized smoothing. Later, Li et al. (2018) improved these bounds, which where further improved by Cohen et al. (2019). Salman et al. (2019) improved the results of Cohen et al. (2019) by performing adversarial training on the smoothed classifier. Wang et al. (2019) similarly inject Gaussian noise into ResNet architectures to increase adversarial robustness.

**Certification of geometric transformations**  Beyond $l_p$ norm bound attacks, a more realistic attacker model includes perturbations such as geometric attacks, i.e., rotations, translations and shearing (Engstrom et al., 2017; Kanbak et al., 2018). Work on certification against geometric attacks was performed by (Pei et al., 2017) using enumeration and (Singh et al., 2019) using abstract interpretation. Neither work scales to large networks and images. Thus, an interesting question is whether we can handle more complex perturbations (e.g., rotations) on larger networks. Towards this, in the rest of the paper, we show a generalization of the smoothing method to this richer class of transformations.

## 3 SMOOTHED CLASSIFIER

In this work we consider parameterized transformations of data points $\boldsymbol{x} \in \mathbb{R}^n$. We denote the *smoothing transformation* with parameter $\boldsymbol{s} \in \mathbb{R}^d$ by $\psi_{\boldsymbol{s}} : \mathbb{R}^n \to \mathbb{R}^n$. Further we denote the *attacker transformation* $\phi_{\delta} : \mathbb{R}^n \to \mathbb{R}^n$ with $\delta \in \mathbb{R}^d$ as the perturbation that the adversary can apply. We require that $\psi$ and $\phi$ compose as $\psi_{\boldsymbol{s}} \circ \phi_{\delta} = \psi_{\boldsymbol{s}+\delta}$. Examples of attacker transformations are image rotations and translations as discussed in Section 4.

Using this notion of transformation we now define a *smoothed classifier*:

**Definition 3.1** (Smoothed Classifier). Given a *base classifier* $f : \mathbb{R}^n \to \{1, \ldots, k\}$ and a transformation $\psi_{\boldsymbol{s}} : \mathbb{R}^n \to \mathbb{R}^n$ with $\boldsymbol{s} \in \mathbb{R}^d$ we define a *smoothed classifier* $g_{\psi}^f : \mathbb{R}^n \to \{1, \ldots, k\}$:

$$g_{\psi}^f(\boldsymbol{x}) = \arg\max_c \mathbb{P}_{\boldsymbol{s}} \left( f(\psi_{\boldsymbol{s}}(\boldsymbol{x})) = c \right), \qquad \boldsymbol{s} \sim \mathcal{N}(0, \Sigma)$$

for a covariance matrix $\Sigma$ (symmetric positive definite matrix).

---

**Algorithm 1** for prediction

  *# evaluate $g$ at $x$*
  **function** PREDICT($f$, $\Sigma$, $x$, $n$, $\alpha$, $\psi$)
    counts $\leftarrow$ SAMPLE($f$, $x$, $n$, $\Sigma$, $\psi$)
    $\hat{c}_A, \hat{c}_B \leftarrow$ top two indices in counts
    $n_A, n_B \leftarrow$ counts$[\hat{c}_A]$, counts$[\hat{c}_B]$
    **if** PVALUETEST($n_A$, $n_A + n_B$, 0.5) $\leq \alpha$
  **return** $\hat{c}_A$
    **else return** ABSTAIN

**Algorithm 2** for certification

  *# certify the robustness of $g$ around $x$*
  **function** CERTIFY($f$, $\Sigma$, $x$, $n_0$, $n$, $\alpha$, $\psi$)
    counts0 $\leftarrow$ SAMPLE($f$, $x$, $n_0$, $\Sigma$, $\psi$)
    $\hat{c}_A \leftarrow$ top index in counts0
    counts $\leftarrow$ SAMPLE($f$, $x$, $n$, $\Sigma$, $\psi$)
    $\underline{p_A} \leftarrow$ LBOUND(counts$[\hat{c}_A]$, $n$, $1 - \alpha$)
    **if** $\underline{p_A} > \frac{1}{2}$ **return** $\hat{c}_A$ and $R$
    **else return** ABSTAIN

---

As it is usually clear from the context, we drop sub- and super- script and just write $g$.

Given this definition we state our main theorem on the robustness of the smoothed classifier w.r.t attacks $\phi$. This is an adapted version of the main theorem presented in Cohen et al. (2019):

**Theorem 3.2.** *We assume a fixed but arbitrary data point $\boldsymbol{x} \in \mathbb{R}^n$, base classifier $f : \mathbb{R}^n \to \{1, \ldots, k\}$ and transformation $\psi_{\boldsymbol{s}} : \mathbb{R}^n \to \mathbb{R}^n$. Further we allow an attacker to choose $\delta$ and apply $\phi_\delta(x)$, s.t. $\psi_{\boldsymbol{s}} \circ \phi_\delta = \psi_{\boldsymbol{s}+\delta}$.*

*Then: If $\mathbb{P}(f(\phi_{\boldsymbol{s}}(x) = c_A) = p_A \geq \underline{p_A} \geq \overline{p_B} = p_B = \max_{c_B \neq c_A} \mathbb{P}(f(\phi_{\boldsymbol{s}}(x) = c_B)$ then $g(\psi_\delta(x)) = c_A$ for all $\delta$ such that $\sqrt{\delta^T \Sigma^{-1} \delta} < \frac{\Phi^{-1}(\underline{p_A}) - \Phi^{-1}(\overline{p_B})}{2}$.*

*Proof.* The proof is similar to the one given by Cohen et al. (2019) and is stated in Appendix A. $\square$

Throughout the paper we refer to $\frac{\Phi^{-1}(\underline{p_A}) - \Phi^{-1}(\overline{p_B})}{2}$ as the certification radius $R$.

The key differences to Cohen et al. (2019) are: (i) we allow general transformations $\psi$ rather than only additive noise, and (ii) we consider a full covariance matrix $\Sigma$ rather than an isotropic one. Specifically for $\psi_{\boldsymbol{s}}(\boldsymbol{x}) = \boldsymbol{x} + \boldsymbol{s}$ and $\Sigma = \sigma^2 I_n$, our theorem exactly recovers the statement from Cohen et al. (2019). These two additions enable us to tackle many interesting transformations. Due to the similarity of Theorem 3.2 to their main theorem we can also use very similar versions of their algorithms, shown in Algorithms 1 and 2. Theorem 3.2 is a deterministic statement, as the only involved random variable $\boldsymbol{s}$ is integrated out. However in practice, the theorem only holds with a certain probability as we have finite amount of samples to estimate $p_A$ (or $\underline{p_A}$) and $p_B$ (or $\overline{p_B}$). The function SAMPLE($f$, $x$, $n$, $\Sigma$, $\psi$) returns $n$ samples of $f(\psi_{\boldsymbol{s}}(\boldsymbol{x}))$, $s \sim \mathcal{N}(0, \Sigma)$. These samples are then used in a statistical test to determine $\underline{p_A}$ or test if it is $> \frac{1}{2}$ with certainty $1 - \alpha$ for a given $\alpha$. In both algorithms $\overline{p_B} = 1 - \underline{p_A}$. The probability that Algorithm 1 returns a class other than $g(\boldsymbol{x})$ is at most $\alpha$ and with probability of at least $1 - \alpha$ Algorithm 2 does not abstain.

## 4 EXAMPLE PERTURBATIONS

We now discuss several practical perturbations $\psi$ which are important and can be handled by our generalization. Specifically, we will consider low parameter image transformations such as changes in contrast and brightening as well as geometric transformations such as rotation and translation, which require special attention to deal with interpolation. Further, we describe low parameter audio transformations such as changes in volume or pitch.

### 4.1 IMAGE PERTURBATIONS & INTERPOLATIONS

**Interpolation** A major issue with image transformations such as rotation and translation (for non-integer offsets) is interpolation. A pixel in the transformed image maps to a location in the original image (not necessarily on the pixel grid) where the pixel value is taken from. To obtain the value, we need to interpolate it. Typical interpolations for images are nearest-neighbor interpolation, bilinear

interpolation and bicubic interpolation. Further, if the transformation moves a coordinate outside of the original image, the pixel value is commonly set to $(0,0,0)^T$ (black).

We explore the issues created by interpolation using the example of rotation, the same method applies to other geometric transformations. We denote the rotation and subsequent interpolation of an image $\boldsymbol{x} \in \mathbb{R}^n$ by an an angle $\beta$ as $\text{rotate}_\beta(\boldsymbol{x})$. In general $\text{rotate}_\beta \circ \text{rotate}_\gamma \neq \text{rotate}_{\beta+\gamma}$ due to interpolation. Thus, if we choose $\psi_\beta := \text{rotate}_\beta$, this does not compose with an angle $\gamma$ chosen by an attacker $\phi_\gamma$. To formalize this problem we write $(\text{rotate}_\beta \circ \text{rotate}_\gamma)(\boldsymbol{x}) = \text{rotate}_{\beta+\gamma}(\boldsymbol{x}) + \epsilon_{\beta,\gamma}(\boldsymbol{x})$ where $\epsilon_{\beta,\gamma}(\boldsymbol{x})$ denotes the interpolation error.

**Modeling Interpolation**  We address this by modeling the composition of two rotations as a single rotation together with addition of $l_2$ noise. The smoothing operation against an attacker utilizing rotation is $\psi_{\boldsymbol{s}}(\boldsymbol{x}) = \psi_{\left(\begin{smallmatrix} \beta \\ \eta \end{smallmatrix}\right)}(\boldsymbol{x}) := \text{rotate}_\beta(\boldsymbol{x}) + \eta$. For ease of notation we write $\psi_{\beta,\eta}$.

We will let the attacker choose $\delta = \left(\begin{smallmatrix} \gamma \\ \boldsymbol{0} \end{smallmatrix}\right)$ but in truth apply $\hat{\delta} = \left(\begin{smallmatrix} \gamma \\ \omega \end{smallmatrix}\right)$ for a $\omega$ yet to be determined to offset the interpolation error. Thus we can model the overall attacking and smoothing process as:

$$(\psi_{\beta,\eta} \circ \phi_{\gamma,\omega})(\boldsymbol{x}) = \psi_{\beta+\gamma,\eta+\omega}(\boldsymbol{x})$$
$$= \text{rotate}_{\beta+\gamma}(\boldsymbol{x}) + \eta + \omega = \text{rotate}_\beta \circ \text{rotate}_\gamma(\boldsymbol{x}) - \epsilon_{\beta,\gamma}(\boldsymbol{x}) + \eta + \omega. \tag{1}$$

Now choosing $\omega = \epsilon_{\beta,\gamma}(\boldsymbol{x})$ we can get rid of the introduced interpolation errors. Thus, our overall transformation $\psi_{\beta,\eta} \circ \phi_{\gamma,\omega}$ recovers the semantic of $\psi_{\beta,\eta} \circ \phi_{\gamma,\boldsymbol{0}}$ without interpolation. Instantiating Theorem 3.2 with this transformation and $\Sigma = \left(\begin{smallmatrix} \sigma_r^2 & \boldsymbol{0} \\ \boldsymbol{0} & \sigma_i^2 I_n \end{smallmatrix}\right)$, where $I_n$ denotes the $n \times n$ Identity matrix, we obtain the bound:

$$\sqrt{\hat{\delta}^T \Sigma^{-1} \hat{\delta}} < R \xRightarrow{R>0} \hat{\delta}^T \Sigma^{-1} \hat{\delta} < R^2$$
$$\implies \sigma_r^{-2}\gamma^2 + \sigma_i^{-2}\hat{\omega}^T\omega < R^2 \implies \gamma^2 < \sigma_r^2 \left(R^2 - \sigma_i^{-2}\|\epsilon_{\beta,\gamma}(\boldsymbol{x})\|_2^2\right)$$
$$\implies |\gamma| < \sigma_r \sqrt{R^2 - \sigma_i^{-2}E^2} \tag{2}$$

with $E$ upper bounding $\|\epsilon_{\beta,\gamma}(\boldsymbol{x})\|_2 \leq E$. In the first step we also need to assume that $R > 0$. When $R$ is negative (or term under the root is negative) we take the bound $|\gamma| = 0$.

**Approach**  Using Eq. (2), we can derive an algorithm that certifies a range of rotation angles and extend CERTIFY to return $|\gamma|$ rather than $R$. There are four ways one can proceed here:

*Per-Input Global Optimization:* If we are given a fixed image $\boldsymbol{x}$ and a range $[\gamma_{\min}, \gamma_{\max}]$ we can calculate the worst case interpolation error by performing global optimization over $\gamma \in [\gamma_{\min}, \gamma_{\max}]$ and $\beta \in [-180, 180]$. In CERTIFY we would first calculate $E$ before proceeding with the rest of the procedure. This procedure can be improved by only considering $\beta \in [-3\sigma_r, 3\sigma_r]$, which likely simplifies the optimization problem and still covers 99.7% of possible angles.

*Per-Input Sampling:* Similar to the global optimization approach, we can, for a given $\boldsymbol{x}$ and a range $[\gamma_{\min}, \gamma_{\max}]$ sample different interpolation errors and obtain a probabilistic guarantee on $E$.

*Training Set Global Optimization:* If we assume that our $\boldsymbol{x}$ will come from the data distribution we can already pre-compute the maximal $E$ over the training set.

*Training Set Sampling:* Again assuming that $\boldsymbol{x}$ follows the data distribution we can sample different $E$ over the training set and rotation angles and take the maximum.

There are two major downsides here: (i) the global maximum of the error norm might be quite large even though the majority of values is actually quite small and (ii) approaches based on the training set or sampling might not find the correct maxima and only provide probabilistic guarantees.

In practice (see Fig. 2; discussed later) most error norms are small under certain assumptions. However, some are vast outliers. This effectively rules out taking the maximum of $E$ (either for specific $\boldsymbol{x}$ or over the whole dataset) as it becomes essentially impossible to obtain a bound in Eq. (2). This rules out taking the global maxima of the error as $E$.

Further, we need to consider that sampling many interpolation errors for a specific image can be done but is expensive. Rotating an $m \times l$ color image with bilinear interpolation requires around

$74 \cdot l \cdot m$ floating point operations which quickly becomes a bottleneck. This rules out creating a new set of samples for every image.

For these two reasons we see computing a probabilistic bound $\mathbb{P}(E \leq t) \leq \varepsilon$ over the training set offline as the only viable way. We treat each sample as a boolean random variable and obtain the Clopper-Pearson upperbound (Clopper & Pearson, 1934) $\overline{\varepsilon}$ with confidence $1 - \rho$.

**Probabilistic Guarantees**   Assume that in CERTIFY or PREDICT we observe $\tilde{n}_A$ and $\tilde{n}_B$ examples of classes $c_A$ and $c_B$ when taking $n = \tilde{n}_A + \tilde{n}_B$ samples. As in Cohen et al. (2019), we can estimate $\tilde{p}_A$, the probability that we observe $c_A$, and lower bound it by $\underline{\tilde{p}_A}$ with confidence $1 - \alpha$.

Due to our probabilistic guarantee on $E$, there are samples that can not be trusted as Eq. (2) and subsequently the bound on $\delta$ in Theorem 3.2 is not satisfied. When estimating $\underline{p_A}$ and $\overline{p_B}$ we must account for these bad samples. Specifically we can assume that in the worst case all of these samples would have counted towards class $c_A$ and construct a conservative lower bound. We can model the real chance of observing clean samples from $c_A$, denoted $p_A$ by using the union bound:

$$\tilde{p}_A = \mathbb{P}(\text{sample from } c_A \vee \text{sample is bad}) \leq \mathbb{P}(\text{sample from } c_A) + \mathbb{P}(\text{sample is bad})$$
$$\implies \tilde{p}_A \leq p_A + \varepsilon$$
$$\implies \underline{p_A} \geq \underline{\tilde{p}_A} - \overline{\varepsilon} \tag{3}$$

Since we had confidence $1 - \alpha$ and $1 - \rho$ for $\underline{\tilde{p}_A}$ and $\overline{\varepsilon}$ the confidence $\underline{p_A}$ for $1 - \alpha - \rho$. In Cohen et al. (2019) the authors note that the probability that PREDICT returns a wrong class and that CERTIFY abstains are both at most $\alpha$, when $\underline{p_A}$ was estimated with confidence $1 - \alpha$. The same applies to our probabilistic approach, but the failure rate becomes $\alpha + \rho$. Fig. 1 shows the overall bound on $|\gamma|$.

Finally, as we obtained our probabilistic guarantee on $E$ with sampling from $\gamma \in \mathcal{U}([-\gamma_{\min}, \gamma_{\max}])$, we need – in order to be sound – mention that we do not guarantee robustness for all $\gamma$ covered by Theorem 3.2, but only those intersecting with $[-\gamma_{\min}, \gamma_{\max}]$.

## 4.2   INTERPOLATION-FREE PIXEL TRANSFORMATION

While we mainly focus on dealing with image perturbations that introduce interpolation error, there are many that do not. A simple class here are perturbations that are additive operations in some color space (RGB, HSV, HSL, etc.). For example, to model *brightness* we consider the perturbation $\psi_\beta(\boldsymbol{x}) = \boldsymbol{x} + \beta \cdot \mathbf{1}$ in RGB space where $\mathbf{1}$ is the vector of ones with the same dimensions as the image $\boldsymbol{x}$. The equation can easily be adapted to have different $\beta$ per color channel or even per region of the image. We can also model multiplicative changes, by bounding them in the $\exp$ domain. As an example, we consider changes in contrast, $\psi_\beta(\boldsymbol{x}) = 128 + e^\beta(\boldsymbol{x} - 128)$

$$\psi_\beta \circ \phi_\gamma(\boldsymbol{x}) = 128 + e^\beta \cdot e^\gamma \cdot (\boldsymbol{x} - 128) = 128 + e^{\beta+\gamma} \cdot (\boldsymbol{x} - 128) = \psi_{\beta+\gamma}(\boldsymbol{x})$$

However we can not combine additive and multiplicative perturbations into linear transformations without additional assumptions about the order in which perturbations are applied.

The perturbation of the attacker is likely applied to the image in integer space (i.e., each pixel intensity value is an integer in $\{0, \ldots, 255\}$). Thus after applying additive or multiplicative scaling the value might be rounded. Hence, to show that perturbations with value $|\beta|$ are safe, we have to verify that in fact $|\beta + 1|$ is safe (depending on the exact rounding behavior adding $\frac{1}{2}$ might be sufficient). For multiplicative changes, a corresponding safety margin can be calculated.

## 4.3   AUDIO PERTURBATIONS

The domain of audio data has similar challenges as the one for image processing. As before, we handle low parameter attacks like shifts in pitch or changes in volume:

**Volume**   The volume of an audio signal can be changed by multiplying the signal with a constant. The smoothing operation is $\psi_\beta(\boldsymbol{x}) := e^\beta \cdot \boldsymbol{x}$ and similarly the adversary operation is $\phi_\gamma(\boldsymbol{x}) := e^\gamma \cdot \boldsymbol{x}$. We see that the composition is additive in the parameter space:

$$\psi_\beta \circ \phi_\gamma(\boldsymbol{x}) = e^\beta \cdot e^\gamma \cdot \boldsymbol{x} = e^{\beta+\gamma} \cdot \boldsymbol{x} = \psi_{\beta+\gamma}(\boldsymbol{x}),$$

where we clip values exceeding the range of the values of x back when needed.

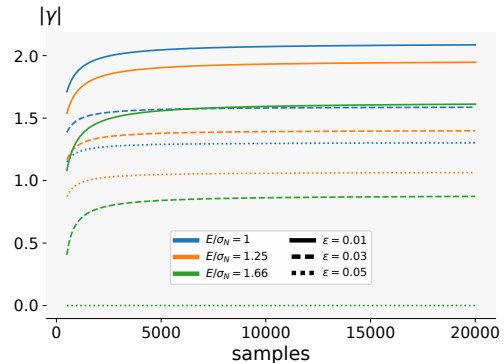
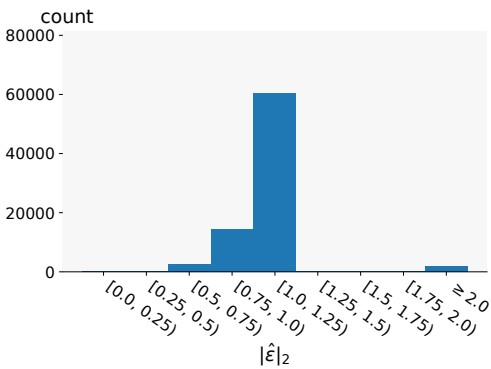

Figure 1: Bound for $|\gamma|$ in the case where the base classifier $f$ returns the class $c_A$ for all samples for $\sigma_r = 1$. Different values of $\sigma_r$ scale the bound multiplicatively.

Figure 2: Histogram of $\|\hat{\epsilon}_{\beta,\gamma}(\boldsymbol{x})\|_2$ for bilinear interpolation after preprocessing for $\boldsymbol{x}$ sampled from the training set, s.t. its shorter side is at least 2000 pixels.

**Pitch shifts**   We let DFT denote the discrete Fourier transform and IDFT is its inverse. The transformation that shifts an audio signal by $\beta$ is $\psi_\beta(\boldsymbol{x}) := \text{IDFT}(\beta + \text{DFT}(\boldsymbol{x}))$. Further, for an attacker $\phi_\gamma := \text{IDFT}(\gamma + \text{DFT}(\boldsymbol{x}))$ the composition of these two is again additive in the parameter space:

$$\psi_\beta \circ \phi_\gamma(\boldsymbol{x}) = \text{IDFT}(\beta + \text{DFT}(\text{IDFT}(\gamma + \text{DFT}(\boldsymbol{x})))) = \text{IDFT}(\beta + \gamma + \text{DFT}(\boldsymbol{x})) = \psi_{\beta+\gamma}(\boldsymbol{x}),$$

where we again neglected the frequencies exceeding the frequency range of the applied DFT. Further, $\beta$ and $\gamma$ are rounded to the frequency steps of the DFT, requiring bound similar to Section 4.2.

## 5   EVALUATION

**Setup**   We evaluated our algorithm on a machine with 16 CPU cores running at $3.5\text{GHz}$ and a GeForce RTX 2080 Ti. We run the perturbation on the images in parallel on CPU and evaluate the network on the GPU with a batch size of 64 in PyTorch (Paszke et al., 2017).

**ImageNet Pipeline and Interpolation Error**   In our evaluation we use the ImageNet classification dataset (Russakovsky et al., 2015). While our algorithm works for general image datasets (with a few restrictions, discussed later in this section) we will specifically discuss this dataset as we require the specifics of the image classification pipleine to precisely model and discuss perturbations. The images in the ImageNet training dataset range in size from $20 \times 17$ to $7056 \times 4488$. Some classifiers are adaptive in the input size, but most neural network based methods are fixed to take images of size $224 \times 224$ (Krizhevsky et al., 2012; He et al., 2016). Thus the standard procedure is to first resize the image such that the shorter side has length 256 and then take the $224 \times 224$ center of the image and run the classifier on this image. We denote all of this preprocessing as $\text{preprocess}(\boldsymbol{x})$. A side effect of this preprocessing is that rotation of less than $|\gamma| \leq 8.13$ degrees is applied to an image, then the preprocessed image will not have black corners, as $\tan^{-1}\left(\frac{256-224}{224}\right) = 8.13$ gives the largest angle for which this holds based on the resizing and cropping size.

Further, many image pipeline implementations can perform these perturbations on floating point images or 8-bit-integer-based images. In the integer case the result of the interpolation is rounded to the closest value in $\{0, \dots, 255\}$. In this work we assume the transformations are performed in integer space as this models a more realistic attacker. We let $\tau$ denote the function that maps images from the integer domain to the $[0, 1]$ floating point domain, by dividing by 255. When adding noise, the pixel values might exceed their bounds, thus they are clamped to 0 and 255 or to 0 and 1 respectively, by the clamping function $\text{clamp}$. To facilitate faster training (Simonyan & Zisserman, 2015) the pixel intensities have their mean subtracted and are scaled to have a standard deviation of 1. This is known as normalization ($\text{normalize}$).

In general we can ignore all preprocessing and replace the base classifier $f$ with $f' = f \circ \text{normalize} \circ \text{clamp} \circ \tau \circ \text{preprocess}$, but in this section we need to specifically look at the interactions between this and the smoothing procedure.

Table 1: $\bar{\varepsilon}$ from samples via Clopper Pearson Interval bounds with 99.9% confidence.

|  | interpolation | $\mathbb{P}(E < 1.00)$ | $\mathbb{P}(E < 1.25)$ | $\mathbb{P}(E < 1.50)$ |
|---|---|---|---|---|
| rotation | nearest | 0.51 | 0.71 | 0.83 |
|  | bilinear | 0.21 | 0.96 | 0.97 |
|  | bicubic | 0.09 | 0.96 | 0.96 |
| translation | nearest | 0.56 | 0.55 | 0.60 |
|  | bilinear | 0.21 | 0.99 | 0.99 |
|  | bicubic | 0.07 | 0.90 | 0.99 |

Table 2: Certification results for translation and rotation with 99.8% confidence.

| Transformation | Model | Abstained | Verified | Accurately verified | Correctly verified |
|---|---|---|---|---|---|
| rotation | $S_{0.5}$ | 23 (35%) | 10 (15%) | 8 (12%) | 6 ( 9%) |
| rotation | $S_{1.0}$ | 16 (25%) | 10 (15%) | 5 ( 8%) | 3 ( 5%) |
| translation | $S_{0.5}$ | 15 (23%) | 20 (31%) | 11 (17%) | 9 (14%) |

Table 3: Certification for brightness changes and rotation with 99.9% confidence.

| Model | Abstained | Verified | Accurately Verified | Correctly Verified | $\beta$ |
|---|---|---|---|---|---|
| R | 20 (20%) | 80 (80%) | 70 (70%) | 64 (64%) | $20.37 \pm 14.59$ |
| $S_{0.5}$ | 47 (47%) | 53 (53%) | 47 (47%) | 25 (25%) | $15.65 \pm 12.61$ |

Table 4: Certification for contrast changes and rotation with 99.9% confidence.

| Model | $\sigma$ | Abstained | Verified | Accurately Verified | Correctly Verified | $\beta$ |
|---|---|---|---|---|---|---|
| R | 0.2 | 44 (44%) | 56 (56%) | 49 (49%) | 45 (45%) | $0.21 \pm 0.17$ |
| R | 0.4 | 73 (73%) | 27 (27%) | 22 (22%) | 19 (19%) | $0.27 \pm 0.21$ |
| $S_{0.5}$ | 0.2 | 77 (77%) | 23 (23%) | 14 (14%) | 14 (14%) | $0.15 \pm 0.10$ |
| $S_{0.5}$ | 0.4 | 90 (90%) | 10 (10%) | 6 (6%) | 5 (5%) | $0.23 \pm 0.18$ |

While one would, based on everyday experience assume that interpolation errors are small, the $l_2$ norm can be surprisingly large. For a large sample of images $x$ from the training and pairs of angles $\beta, \gamma$ we measure the norm of the interpolation error $\|\epsilon_{\beta,\gamma}(x)\|$. We observed a mean error of 3.9 with a standard deviation of 3.3 and a maximum of 60.1 A key observation is, that larger images have lower errors. For example, taking a random $250 \times 250$ image $x$ from the ImageNet training set and calculating $\|(\tau \circ \text{preprocess})(x) - (\tau \circ \text{preprocess} \circ \text{rotate}_{-8} \circ \text{rotate}_8)(x)\|_2$ yields a value of 5.5, but still a very similar image. For comparison the largest $l_2$ perturbations considered in provable robustness are around 3.8 (Cohen et al., 2019; Salman et al., 2019).

Scaling the image up by a factor of 4 before performing this routine reduces the error norm already to 1.49. This is not merely an artifact of scaling up the images, but that larger images in general have a quite low error, as can be seen for example in Fig. 2 which shows interpolation errors $\|\hat{\epsilon}\|$ for images such that the shorter side is at least 2000 pixels long.

**Rotation & Translation**  To apply rotation and translation we use the framework developed in Section 4. However, in practice we apply the rotation before any preprocessing (as this is where the real attacker would do), and the error-offsetting $l_2$ noise after, as preprocessing operations such as down-scaling and cropping tend to reduce it. This does not weaken the attacker, but is only a technical detail of the implementation. So specifically we consider a classifier $f'' = f \circ \text{normalize} \circ \text{clamp}$ and a perturbation $\hat{\psi}_{\beta+\gamma,\eta+\omega}(x) = (\tau \circ \text{preprocess} \circ \text{rotate}_\beta \circ \text{rotate}_\gamma(x)) - \hat{\epsilon} + \eta + \hat{\omega}$. where $\hat{\epsilon} = (\tau \circ \text{preprocess} \circ \text{rotate}_\beta \circ \text{rotate}_\gamma)(x) - (\tau \circ \text{preprocess} \circ \text{rotate}_{\beta+\gamma})(x)$.

Replacing the perturbation and $\epsilon$ in Eq. (1) with these ultimately yields to a smaller bound on $E$ in Eq. (2). Due to the size issues outlined before, we consider only images from ImageNet where the shorter side has at least 2000 pixels. Fig. 2 shows the distribution over $\|\hat{\epsilon}\|$ over a sample of these images and different angles $\beta \sim \mathcal{N}(0, 5^2)$, $\gamma \in \mathcal{U}([-8, 8])$. We chose these distributions to model the errors, actually observed during the smoothing procedure: Here we assume an attacker to choose $\gamma \in [-8, 8]$ and $\sigma_r = 5$. Based on these samples we derive $\mathbb{P}(E \le t)$ listed in Table 1. More details, including histograms for translations are given in Fig. 3 in the Appendix.

We now evaluate our algorithm on the images of the ImageNet test set, where the shorter side is at least 2000 pixels, of which there are 65. Since we are largely bound by the $l_2$ robustness of the classifier we use two classifiers from Salman et al. (2019), which were trained to be robust base classifiers for $l_2$ Gaussian smoothing: $S_{0.5}$ denotes a ResNet-50 (He et al., 2016) trained with SMOOTHADV$_{\text{PGD}}$, $\sigma = 0.5$, $\epsilon = 1.0$, which had the best approximate certified test accuracy for a noise levels of 1.0 and 1.5 and $S_{1.0}$ denotes the same model trained with $\sigma = 1.0$. While this has slightly worse performance it was trained with a higher $\sigma$ (robustness to $l_2$ noise, allowing us to use a larger $\sigma_n$. On our data the base classifier $S_{0.5}$ was correct on 36 samples and $S_{1.0}$ on 19. Further, we are using $E = 1.25$, $\sigma_n = 0.75$, $\sigma_r = 5$, $\alpha = 0.001$, $\rho = 0.001$, $n_0 = 100$, $n = 10000$, and bilinear interpolation.

The results are shown in Table 2 where we consider the following evaluation metrics: (i) *Abstained* shows how often the classifier did not return a certified classification, (ii) *Verified* is how often a radius larger than 0 was proven to be correct for a class (note that the verification is approximate with the stated confidence), (iii) *Accurately Verified* shows how many samples could be verified for the same class as the base classifier predicted, and (iv) *Correctly Verified* shows how many samples were verified and also had the correct label to the dataset.

The biggest trade-off in these experiments is choosing $\sigma_n$ as large as possible, to obtain a good bound in Eq. (2). However, at the same time this lowers the accuracy and thus decreases the bound. Interestingly more samples don't necessarily help us as much as they do in Cohen et al. (2019) as our estimate of $\underline{p_A}$ is fundamentally limited by Eq. (3).

For rotation with $S_{0.5}$ the mean $\pm$ standard deviation is $3.24 \pm 0.77$ degrees for the verified examples and $3.56 \pm 0.50$ degrees for $S_{1.0}$. For transformation on $S_{0.5}$, $\| \binom{dx}{dy} \|_2 \le 10.70 \pm 3.60$, where $dx$ and $dy$ denotes the offset in $x$ and $y$ direction respectively. To be sound we would need to limit $dx$ and $dy$ to at most $\pm 2$ as this was our assumption when obtaining $E$, but the distribution for larger changes is similar to the one obtained for $\pm 2$ so an argument for the full range can be made. The average run time to certify rotations with $n = 10000$ is $256.27s$ and for translations $250.02s$.

We conclude that the main bottlenecks are the inequality Eq. (3), the accuracy of the base classifier and its robustness to $l_2$ allowing us to choose larger $\sigma_n$.

**Brightness and Contrast**  In contrast to the perturbations that perform interpolation we do not need to use Eq. (2) but can directly use Theorem 3.2. So for brightness changes and contrast changes we use $\Sigma = \sigma$ (a scalar), and since for this task we don't specifically need an $l_2$ robust network, we use a standard ResNet-50 (He et al., 2016) from PyTorch Torchvision (Paszke et al., 2017), denoted as R, as well as $S_{0.5}$ from before. The results are shown in Tables 3 and 4. For brightness we use $\sigma = 12$ and for contrast $\sigma = 0.2$ as well as $\sigma = 0.4$. For both we use $n_0 = 100$, $n = 10000$ and $\alpha = 0.001$ and observe an average evaluation time for $n = 10000$ of 24.01 s. To be sound w.r.t. integer rounding we need to subtract 1 and 0.0008 from the $\beta$ in Table 3 and Table 4 respectively. The results highlight how much the accuracy and robustness of the base classifier aid verification. R is much more accurate than $S_{0.5}$ allowing us to certify large ranges of brightness changes.

## 6  CONCLUSION

We presented a way to extend the Gaussian Smoothing framework (Cohen et al., 2019) to interesting perturbations in application domains such as image and audio classification. In our evaluation we showed that the approach is applicable to complex tasks such as ImageNet classification, although we believe that improvements to classifiers and application domain specific insights can strongly improve the results. We believe that this work makes Gaussian Smoothing applicable outside the often considered $l_p$-ball and will trigger further work in this direction.

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

## A    PROOF OF THEOREM 3.2

The assumption is

$$\mathbb{P}\left((f \circ \psi_{\boldsymbol{s}})(\boldsymbol{x}) = c\right) = p_c \geq \underline{p_c} \geq \overline{p'_c} \geq p'_c = \mathbb{P}\left((f \circ \psi_{\boldsymbol{s}})(\boldsymbol{x}) = c'\right).$$

By the definition of $g$ we need to show that

$$\mathbb{P}\left((f \circ \psi_{\boldsymbol{s}} \circ \phi_\delta)(\boldsymbol{x}) = c\right) \geq \mathbb{P}\left((f \circ \psi_{\boldsymbol{s}} \circ \phi_\delta)(\boldsymbol{x}) = c'\right).$$

Using $\psi_{\boldsymbol{s}} \circ \phi_\delta = \psi_{\boldsymbol{s}+\delta}$, we get

$$\mathbb{P}\left((f \circ \psi_{\boldsymbol{s}+\delta})(\boldsymbol{x}) = c\right) \geq \mathbb{P}\left((f \circ \psi_{\boldsymbol{s}+\delta})(\boldsymbol{x}) = c'\right).$$

We define the set $A := \{\boldsymbol{z} \mid \delta^T \Sigma^{-1} \boldsymbol{z} \leq \sqrt{\delta^T \Sigma^{-1} \delta} \Phi(\underline{p_c})\}$. We claim that

$$\mathbb{P}(f \circ \psi_{\boldsymbol{s}}(x) = c) \geq \mathbb{P}(\boldsymbol{s} \in A) \tag{4}$$
$$\mathbb{P}(f \circ \psi_{\boldsymbol{s}+\delta}(x) = c) \geq \mathbb{P}(\boldsymbol{s} + \delta \in A) \tag{5}$$

holds. First, we show that Eq. (4) holds.

$$\begin{aligned}
\mathbb{P}(\boldsymbol{s} \in A) &= \mathbb{P}(\delta^T \Sigma^{-1} \boldsymbol{s} \leq \sqrt{\delta^T \Sigma^{-1} \delta} \Phi(\underline{p_c})) \\
&= \mathbb{P}(\delta^T \Sigma^{-1} \mathcal{N}(0, \Sigma) \leq \sqrt{\delta^T \Sigma^{-1} \delta} \Phi(\underline{p_c})) \\
&= \mathbb{P}(\delta^T \sqrt{\Sigma^{-1}} \mathcal{N}(0, \mathbb{1}) \leq \sqrt{\delta^T \Sigma^{-1} \delta} \Phi(\underline{p_c})) \\
&= \mathbb{P}(\mathcal{N}(0, \delta^T \Sigma^{-1} \delta) \leq \sqrt{\delta^T \Sigma^{-1} \delta} \Phi(\underline{p_c})) \\
&= \mathbb{P}(\sqrt{\delta^T \Sigma^{-1} \delta} \mathcal{N}(0, \mathbb{1}) \leq \sqrt{\delta^T \Sigma^{-1} \delta} \Phi(\underline{p_c})) \\
&= \mathbb{P}(\mathcal{N}(0, \mathbb{1}) \leq \Phi(\underline{p_c})) \\
&= \Phi(\Phi^{-1}(\underline{p_c})) \\
&= \underline{p_c}
\end{aligned}$$

Thus Eq. (4) holds. Next we show that Eq. (5) holds

$$\begin{aligned}
&\int_{\mathbb{R}^d} [f \circ \psi_{\boldsymbol{z}} = c] p_{\boldsymbol{s}+\delta}(z) dz - \int_A p_{\boldsymbol{s}+\delta}(z) dz \\
&= \int_{\mathbb{R}^d \backslash A} [f \circ \psi_{\boldsymbol{z}}(x) = c] p_{\boldsymbol{s}+\delta}(z) dz + \int_A ([f \circ \psi_{\boldsymbol{z}}(x) = c] - 1) p_{\boldsymbol{s}+\delta}(z) dz \\
&= \int_{\mathbb{R}^d \backslash A} [f \circ \psi_{\boldsymbol{z}}(x) = c] p_{\boldsymbol{s}+\delta}(z) dz \\
&\quad + \int_A ([f \circ \psi_{\boldsymbol{z}}(x) = c] - [f \circ \psi_{\boldsymbol{z}}(x) = c] - [f \circ \psi_{\boldsymbol{z}}(x) \neq c]) p_{\boldsymbol{s}+\delta}(z) dz \\
&= \int_{\mathbb{R}^d \backslash A} [f \circ \psi_{\boldsymbol{z}}(x) = c] p_{\boldsymbol{s}+\delta}(z) dz - \int_A [f \circ \psi_{\boldsymbol{z}}(x) \neq c] p_{\boldsymbol{s}+\delta}(z) dz \\
&\overset{Lemma\ A.1}{\geq} \int_{\mathbb{R}^d \backslash A} [f \circ \psi_{\boldsymbol{z}}(x) = c] p_{\boldsymbol{s}}(z) dz - \int_A [f \circ \psi_{\boldsymbol{z}}(x) \neq c] p_{\boldsymbol{s}}(z) dz \\
&= \int_{\mathbb{R}^d} [f \circ \psi_{\boldsymbol{z}}(x) = c] p_{\boldsymbol{s}}(z) dz - \int_A p_{\boldsymbol{s}}(z) dz \\
&\overset{Eq.\ (4)}{\geq} 0.
\end{aligned}$$

Thus also Eq. (5) holds.

**Lemma A.1.** *There exists $t > 0$ such that $p_{\boldsymbol{s}+\delta}(z) \leq p_{\boldsymbol{s}}(z) \cdot t$ for all $z \in A$.*

*Proof.*

$$\frac{p_{s+\delta}(z)}{p_s(z)} = \exp\left(-\frac{1}{2}(z-\delta)^T\Sigma^{-1}(z-\delta) + \frac{1}{2}z^T\Sigma^{-1}z\right)$$
$$= \exp\left(-\frac{1}{2}z^T\Sigma^{-1}z + z^T\Sigma^{-1}\delta - \frac{1}{2}\delta^T\Sigma^{-1}\delta + \frac{1}{2}z^T\Sigma^{-1}z\right)$$
$$= \exp\left(z^T\Sigma^{-1}\delta - \frac{1}{2}\delta^T\Sigma^{-1}\delta\right)$$

What is the lowest $t$ if it exists such that $\frac{p_{s+\delta}(z)}{p_s(z)} \le t$?

$$\frac{p_{s+\delta}(z)}{p_s(z)} \le t$$
$$\Leftrightarrow \quad \exp\left(z^T\Sigma^{-1}\delta - \frac{1}{2}\delta^T\Sigma^{-1}\delta\right) \le t$$
$$\Leftrightarrow \quad z^T\Sigma^{-1}\delta - \frac{1}{2}\delta^T\Sigma^{-1}\delta \le \log t$$
$$\Leftrightarrow \quad z^T\Sigma^{-1}\delta \le \log t + \frac{1}{2}\delta^T\Sigma^{-1}\delta$$

Because $z \in A$, we know that

$$z^T\Sigma^{-1}\delta \le \sqrt{\delta^T\Sigma^{-1}\delta}\,\Phi^{-1}(\underline{p_c}).$$

Does there exist a $t$ such that both upper bound coincide? Yes, namely

$$t = \exp\left(\sqrt{\delta^T\Sigma^{-1}\delta}\,\Phi^{-1}(\underline{p_c}) - \frac{1}{2}\delta^T\Sigma^{-1}\delta\right).$$

$\square$

Next, we claim that for $B := \{z \mid \delta^T\Sigma^{-1}z \ge \sqrt{\delta^T\Sigma^{-1}\delta}\,\Phi^{-1}(1-\overline{p_c'})\}$ holds that

$$\mathbb{P}(f \circ \psi_s(x) = c') \le \mathbb{P}(s \in B) \tag{6}$$
$$\mathbb{P}(f \circ \psi_{s+\delta}(x) = c') \le \mathbb{P}(s + \delta \in B) \tag{7}$$

The proof for Eq. (6) and Eq. (7) are analogous to the proofs for Eq. (4) and Eq. (5).

Now we derive the conditions that lead to $\mathbb{P}(s + \delta \in A) > \mathbb{P}(s + \delta \in B)$:

$$\mathbb{P}(s + \delta \in A) = \mathbb{P}\left(\delta^T\Sigma^{-1}(s+\delta) \le \sqrt{\delta^T\Sigma^{-1}\delta}\,\Phi^{-1}(\underline{p_c})\right)$$
$$= \mathbb{P}\left(\delta^T\Sigma^{-1}(\sqrt{\Sigma}\mathcal{N}(0,\mathbb{1}) + \delta) \le \sqrt{\delta^T\Sigma^{-1}\delta}\,\Phi^{-1}(\underline{p_c})\right)$$
$$= \mathbb{P}\left(\delta^T\sqrt{\Sigma^{-1}}\mathcal{N}(0,\mathbb{1}) + \delta^T\Sigma^{-1}\delta \le \sqrt{\delta^T\Sigma^{-1}\delta}\,\Phi^{-1}(\underline{p_c})\right)$$
$$= \mathbb{P}\left(\sqrt{\delta^T\Sigma^{-1}\delta}\mathcal{N}(0,\mathbb{1}) + \delta^T\Sigma^{-1}\delta \le \sqrt{\delta^T\Sigma^{-1}\delta}\,\Phi^{-1}(\underline{p_c})\right)$$
$$= \mathbb{P}\left(\mathcal{N}(0,\mathbb{1}) + \sqrt{\delta^T\Sigma^{-1}\delta} \le \Phi^{-1}(\underline{p_c})\right)$$
$$= \mathbb{P}\left(\mathcal{N}(0,\mathbb{1}) \le \Phi^{-1}(\underline{p_c}) - \sqrt{\delta^T\Sigma^{-1}\delta}\right)$$
$$= \Phi(\Phi^{-1}(\underline{p_c}) - \sqrt{\delta^T\Sigma^{-1}\delta})$$

Similarly, we have

$$\mathbb{P}(s + \delta \in B) = \mathbb{P}\left(\mathcal{N}(0,\mathbb{1}) \ge \Phi^{-1}(1-\overline{p_{c'}}) - \sqrt{\delta^T\Sigma^{-1}\delta}\right)$$
$$= \Phi(\sqrt{\delta^T\Sigma^{-1}\delta} - \Phi^{-1}(1-\overline{p_{c'}}))$$

Thus, we get

$$
\begin{aligned}
\mathbb{P}(\boldsymbol{s}+\delta \in A) \quad &> \mathbb{P}(\boldsymbol{s}+\delta \in B) \\
\Leftrightarrow \quad \Phi(\Phi^{-1}(\underline{p_c}) - \sqrt{\delta^T \Sigma^{-1}\delta}) &> \Phi(\sqrt{\delta^T \Sigma^{-1}\delta} - \Phi^{-1}(1-\overline{p_{c'}})) \\
\Leftrightarrow \quad \Phi^{-1}(\underline{p_c}) - \sqrt{\delta^T \Sigma^{-1}\delta} \quad &> \sqrt{\delta^T \Sigma^{-1}\delta} - \Phi^{-1}(1-\overline{p_{c'}}) \\
\Leftrightarrow \quad \Phi^{-1}(\underline{p_c}) + \Phi^{-1}(1-\overline{p_{c'}}) &> 2\sqrt{\delta^T \Sigma^{-1}\delta} \\
\Leftrightarrow \quad \tfrac{1}{2}(\Phi^{-1}(\underline{p_c}) - \Phi^{-1}(\overline{p_{c'}})) \quad &> \sqrt{\delta^T \Sigma^{-1}\delta}.
\end{aligned}
$$

## B   FURTHER FIGURES

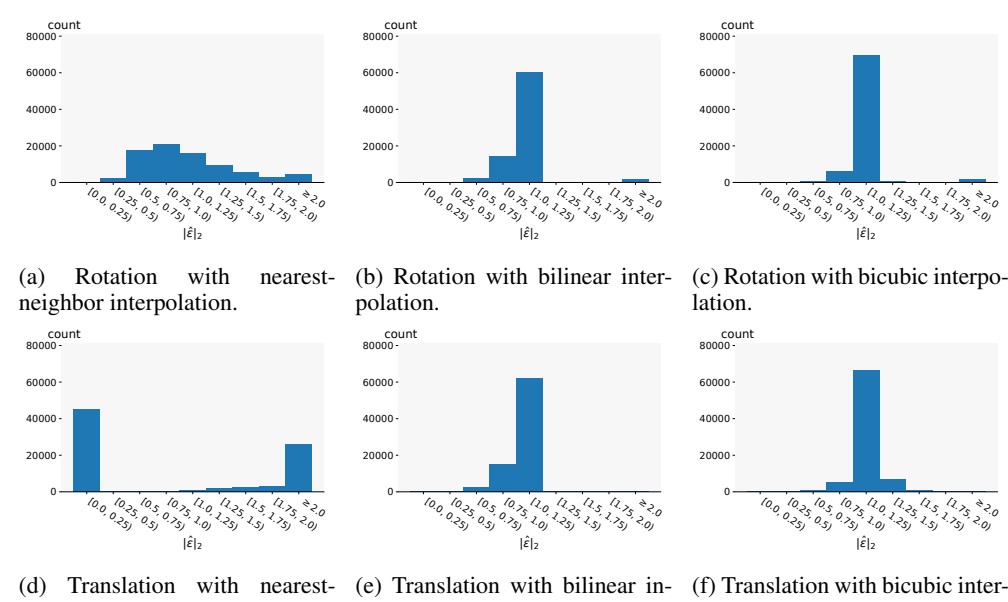

(a) Rotation with nearest-neighbor interpolation.

(b) Rotation with bilinear interpolation.

(c) Rotation with bicubic interpolation.

(d) Translation with nearest-neighbor interpolation.

(e) Translation with bilinear interpolation.

(f) Translation with bicubic interpolation.

Figure 3: Histogram of $\|\hat{\epsilon}_{\beta,\gamma}(\boldsymbol{x})\|_2$ for translation and roation after preprocessing for $\beta \sim \mathcal{N}(0, 5^2)$, $\gamma \in \mathcal{U}([-8, 8])$ and $\boldsymbol{x}$ from the training set conditioned on the fact that the shorter side of $\boldsymbol{x}$ is at least 2000 pixels.

