# OpenReview forum: "Statistical Verification of General Perturbations by Gaussian Smoothing"
_ICLR.cc/2020/Conference — Reject_

### Official Review · AnonReviewer4 · 2019-10-21
**Official Blind Review #4**

**Rating:** 3

**Review:**

This paper applied the framework of randomized smoothing classifier proposed by Cohen et al. to certify the adversarial attacks other than pixel change, including image rotating, brightness change in RGB space, Volume and pitch shifts for audio.

Certifying general adversarial attack is an interesting direction. However, I do not believe this paper is qualified for publishing in ICLR. Below please find my comments:

Different from change-pixel attack, the certification for rotation/brightness change in image classification, volume and pitch change in audio perturbations is much easier. The studied perturbation can be parameterized by a very low dimension vector (i.e. one dimension (angle and volume) for image rotation and volume change, 3 dimensions (brightness for each channel) for brightness perturbation). To certify this attack, given a base classifier, we can simply do, for example, grid search, on the low dimensional space to find the worst case with very good accuracy. And it should be able to give much better than the randomized smoothing method.

One contribution of this paper is in that: Theorem 3.2 is valid for random smoothing using gaussian with general covariance matrix. However, the effectiveness of using a general covariance matrix is not studied. In the experiment section, I find only isotropic ones are used.

**Experience Assessment:**

I have read many papers in this area.

**Review Assessment: Checking Correctness Of Derivations And Theory:**

I carefully checked the derivations and theory.

**Review Assessment: Checking Correctness Of Experiments:**

I assessed the sensibility of the experiments.

**Review Assessment: Thoroughness In Paper Reading:**

I read the paper at least twice and used my best judgement in assessing the paper.

---

> ### Author Response · Authors · 2019-11-14
> **Response to Review #4**
>
> > Non-isotropic covariance matrices
>
> We have addressed this point in our response to all reviewers.
>
> > To certify this attack, given a base classifier, we can simply do, for example, grid search, on the low dimensional space to find the worst case with very good accuracy. And it should be able to give much better than the randomized smoothing method.
>
> With interpolation scheme such as linear and bilinear interpolation, every legal floating-point parameter would produce a (potentially) different interpolation. Thus enumeration (even though theoretically possible as floats are countable) is infeasible. For the case of nearest-neighbor interpolation Pei et al. [1] investigated an enumerative approach.
>
> [1] Towards Practical Verification of Machine Learning: The Case of Computer Vision Systems, Pei et al., https://arxiv.org/abs/1712.01785

---

### Official Review · AnonReviewer2 · 2019-10-22
**Official Blind Review #2**

**Rating:** 3

**Review:**

In this paper, the authors generalize the randomized smoothing type of robustness certification to handle many types of attacks beyond norm-based attacks, e.g., geometric perturbation, volume change, pitch shifts on audio data. At the core of the proposed generalization is using some interpolation which I think is quite straightforward.

1. How about the scenario when both rotation and translation exist?

2. In experiments, can the authors consider the accuracy of the certified model under the adversarial setting, i.e., find the transformation that makes the deep nets performs worst?

3. Besides randomized smoothing on the input images, recently Wang et al showed that randomize the deep nets can
also improve the deep nets and they gave it a nice theoretical interpretation. Here is the reference: Bao Wang, Binjie Yuan, Zuoqiang Shi, Stanley J. Osher. ResNets Ensemble via the Feynman-Kac Formalism to Improve Natural and Robust Accuracies, arXiv:1811.10745, NeurIPS, 2019

4. Interpolation idea has been used in improving robustness of deep nets, for example:
1). Bao Wang, Xiyang Luo, Zhen Li, Wei Zhu, Zuoqiang Shi, Stanley J. Osher. Deep Neural Nets with Interpolating Function as Output Activation, NeurIPS, 2018
2). Bao Wang, Alex T. Lin, Zuoqiang Shi, Wei Zhu, Penghang Yin, Andrea L. Bertozzi, Stanley J. Osher. Adversarial Defense via Data Dependent Activation Function and Total Variation Minimization, arXiv:1809.08516, 2018
3). B. Wang, S. Osher. Graph Interpolating Activation Improves Both Natural and Robust Accuracies in Data-Efficient Deep Learning, arXiv:1907.06800

In sum, this paper studies the problem of certifying a broad class of adversarial attacks which is decent, however, the novelty is quite limited. Please address my questions during rebuttal.

**Experience Assessment:**

I have published one or two papers in this area.

**Review Assessment: Checking Correctness Of Derivations And Theory:**

I assessed the sensibility of the derivations and theory.

**Review Assessment: Checking Correctness Of Experiments:**

I assessed the sensibility of the experiments.

**Review Assessment: Thoroughness In Paper Reading:**

I read the paper at least twice and used my best judgement in assessing the paper.

---

> ### Author Response · Authors · 2019-11-14
> **Response to Review #2**
>
> > How about the scenario when both rotation and translation exist?
>
> We currently do not support this as a rotation concatenated with translation (and interpolation artifacts) does not commute with translation and rotation. We hope to address this issue in future work.
>
> > In experiments, can the authors consider the accuracy of the certified model under the adversarial setting, i.e., find the transformation that makes the deep nets performs worst?
>
> Similar to Salman et al. [3] we could write an optimization problem to reflect this. Depending on the transformation the problem might be (piece wise) differentiable in the argument. However for rotations, prior research [1] found that randomly sampling perturbations is more effective. We will strive to include experimental results for this in the next version of the paper.
>
> > Randomized Nets and Manifold Interpolation by Wang
>
> We were unaware of this line of work and thank the reviewer of the pointer. Similar to Cohen et al. [2] this addresses norm-ball perturbations. We added it to our related work.
>
> [1] Exploring the Landscape of Spatial Robustness, Engstrom et al., ICML 2019, https://arxiv.org/abs/1712.02779
> [2] Certified Adversarial Robustness via Randomized Smoothing, Cohen et al. ICML 2019, https://arxiv.org/abs/1902.02918
> [3] Provably Robust Deep Learning via Adversarially Trained Smoothed Classifiers, Salman et al., https://arxiv.org/abs/1906.04584

---

### Official Review · AnonReviewer3 · 2019-10-23
**Official Blind Review #3**

**Rating:** 3

**Review:**

Summary:

This paper introduces a new smoothing algorithm which produces classifiers with certified robustness against adversarial perturbations. In particular, the authors are interested in settings where the input vectors need not be robust simply to \ell_2 perturbations, for which previous authors (e.g., Cohen et al.) have already introduced a "randomized smoothing" classifier which possesses the desired properties. Instead, motivated by the desire to certify robustness to other families of transformations, the authors propose to introduce a smoothing transform that operates in an additive way on the underlying parameter space (rather than on the raw input vectors, as in the case of \ell_2 randomized smoothing). The authors provide details for a few examples from image and audio data processing, and then show experimental results on ImageNet data.

Comments:

My assessment of this paper is that the level of novelty is relatively low. On the technical side, I do not believe this paper makes any significant contributions. The authors themselves note that the proof of the main result, Theorem 3.2, closely parallels the proof of Cohen et al. Algorithmically, the proposals of the authors are also minor tweaks of the algorithms in Cohen et al., since the types of perturbations considered in this paper are essentially single-parameter problems. (As a question: The authors prove Theorem 3.2 where the randomized smoothing operation adds possible nonisotropic Gaussian noise in the parameter space. However, it seems that in the examples, the noise added is isotropic -- it is not clear how the authors are choosing \sigma_i in relation to \sigma_r for the interpolation example. Can the authors clarify this, and give examples where smoothing with nonisotropic noise would actually be useful?)

The only real tweak that the authors introduce when presenting their smoothing algorithm is to introduce a small correction term in order to deal with interpolation/rounding errors. However, I do not think that this change is particularly novel.

As a final comment, I think it would have been interesting for the authors to include some simulations with audio data as well, since this constitutes one of the main motivating applications discussed by the authors.

**Experience Assessment:**

I have read many papers in this area.

**Review Assessment: Checking Correctness Of Derivations And Theory:**

I assessed the sensibility of the derivations and theory.

**Review Assessment: Checking Correctness Of Experiments:**

I assessed the sensibility of the experiments.

**Review Assessment: Thoroughness In Paper Reading:**

I read the paper at least twice and used my best judgement in assessing the paper.

---

> ### Author Response · Authors · 2019-11-14
> **Response to Review #3**
>
> > Concerns about contribution and novelty of dealing with interpolation noise & Non-isotropic Sigma
>
> We have addressed these points in our response to all reviewers.
>
> > How are $\sigma_i$ vs $\sigma_r$ chosen?
>
> To certify for large angles $\gamma$, in Equation (2) we want as large values for $\sigma_i$ and $\sigma_r$ as possible. The largest possible values for $\sigma_i$ are dictated by the current results in smoothing for L2-perturbations. By Table 6 in Appendix G of Salman et al. [1] we choose 0.75, as this provides a good trade-off between robustness and accuracy.
> The choice of $\sigma_r$ matters less than the choice of as $\sigma_i$ large values don’t decrease the accuracy as much. As we are only certifying values of $\gamma$ up to $\pm 8$ degrees, $\sigma_r = 5$ was a good trade-off between accuracy and certifiability.
>
> > Audio experiments missing
>
> We will add audio experiments in the next iteration of the paper.
>
> [1] Provably Robust Deep Learning via Adversarially Trained Smoothed Classifiers, Salman et al., 2019, https://arxiv.org/abs/1906.04584

---

### Author Response · Authors · 2019-11-14
**Common Response**

We thank all reviewers for their comments and valuable feedback. We have addressed the individual reviewer comments below and now address common concerns here.

> Impact of work

All reviewers have questioned the novelty of the presented work. We would like to address this by pointing out that the problem of these general perturbations is interesting to the research community [1 - 4]. Dealing with interpolation artifacts is a key challenge in achieving robustness to geometric perturbations of images in our and other works [3, 4]. We believe our approach is a suitable and non-straight-forward way to approach the problem.

> Non-isotropic covariance matrices

Cohen et al.’s [5] original approach requires only a single (co)variance that is the same for all dimensions. For the perturbations discussed in this work (specifically in section 4) a diagonal covariance matrix is required. Since Theorem 3.2 is the same of diagonal and full covariance matrices we stated it is such. However. for perturbations parameterized by multiple (non-noise) parameters the off-diagonal entries can be used to encode dependencies between the parameters, such as the translation in the x direction and the translation in the y direction. While the diagonal version can proof safety for L2-balls in the parameter space, the off-diagonal entries admits general ellipsoids.


[1] Exploring the Landscape of Spatial Robustness, Engstrom et al., ICML 2019, https://arxiv.org/abs/1712.02779
[2] Geometric Robustness of Deep Networks: Analysis and Improvement, Kanbak et al., CVPR 2018, https://arxiv.org/abs/1711.09115
[3] Certifying Geometric Robustness of Neural Networks, Balunovic et al., to appear in NIPS 2019, https://www.sri.inf.ethz.ch/publications/balunovic2019geometric
[4] Towards Practical Verification of Machine Learning: The Case of Computer Vision Systems, Pei et al., https://arxiv.org/abs/1712.01785
[5] Certified Adversarial Robustness via Randomized Smoothing, Cohen et al. ICML 2019, https://arxiv.org/abs/1902.02918

---

### Author Response · Authors · 2021-01-04
**Published Version**

An updated version of this paper was accepted at NeurIPS 2020 and can be found here: https://arxiv.org/abs/2002.12463

---

### Decision · Program_Chairs · 2019-12-19

**Decision:**

Reject

**Comment:**

This paper proposes a smoothing-based certification against various forms of transformations, such as  rotations, translations. The reviewers have concerns on the novelty of the work and several technical issues. The authors have made efforts to address some of issues, but the work may still significantly benefit from a throughout improvement in both presentation and technical contribution.